# Operational feasibility of the ultra-portable digital X-rays with Computer-Aided Detection (CAD) for community active case finding for TB in Nigeria: Health care workers and client's perspectives

Sani Useni[1], Rupert Eneogu[2], Austin Ihesie[2,3], Abiola Alege[4], Aderonke Agbaje[5], Bethrand Odume[1], Debby Nongo[2], Eze Chukwu[1], Jamiu Olabamiji[5], Omosalewa Oyelaran[2], Charles Ohikhuai[6], Zhi Zhen Qin[7], Rachael Barrett[7], Olufunmilayo Omosebi[8], Chukwuma Anyaike[3,8], Atana Ewa[3,9,10]*

1 Programs, KNCV, Abuja, Nigeria, 2 HIVAIDS &TB Office, United States Agency for International Development, Abuja, Nigeria, 3 National Tuberculosis Research Task Team, Abuja, Nigeria  4 Programs, Society for Family Health, Abuja, Nigeria, 5 Programs, Institute of Human Virology, Abuja, Nigeria, 6 Viamo Technologies Limited, Abuja, Nigeria, 7 Digital Health, Stop TB Partnership, Geneva, Switzerland, 8 Department of Public Health, Federal Ministry of Health, Abuja, Nigeria, 9 Department of Paediatrics, University of Calabar, Calabar, Nigeria, 10 Department of Paediatrics, University of Calabar Teaching Hospital, Calabar, Nigeria

* atanaewa@yahoo.com

## Abstract

Nigeria received 10 Ultra-Portable digital X-rays (UPDX) with Computer-Aided Detection (CAD) from the Stop TB Partnership, as part of the USAID-funded introducing New Tools Project (iNTP). The UPDX machines (Delft Light systems) were deployed to 8 states for TB screening and triaging algorithms for early TB detection. This study sought to explore the perspectives of health workers and clients on the operational feasibility and ease-of-use of the UPDX with CAD, with a view to determining the acceptability, enablers and barriers to effective use and exploring the perceived ease-of-use by the end-users. This is a cross-sectional descriptive study, conducted between April and June 2023, using mixed quantitative and qualitative methods to determine the feasibility, acceptability, and ease-of-use of the UPDX with CAD in Nigeria. Purposive sampling was done for 57 respondents made up of radiographers, radiologists, key informants, TBLS/DOT nurses and clients. All aged from 20-60 years. They reported ease-of-use and access to screening, portability, availability in hard-to-reach areas, usefulness for mass screening in communities, comparable quality of x-ray with fixed x-ray, rapid results and had challenges with equipment/ implementation costs, fears of irradiation, lack of collaboration with other programs and inability to use UPDX-with-CAD on children < 4 years. All (100%) strongly agreed that the UPDX/accessories were easy to set up, considered themselves proficient with operating the UPDX and to a lesser extent agreed that the programs and software are user-friendly and easy to set up. They however disagreed that the device is

**Data availability statement:** All data can be found in the manuscript and Supporting Information files.

**Funding:** The study was funded by the United States Agency for International Development (USAID) through the Stop TB Partnership introducing New Tools Project (iNTP). The views and opinions expressed in this paper are those of the authors and not necessarily the views and opinions of the United States Agency for International Development.

**Competing interests:** The authors have declared that no competing interests exist.

portable enough for one person to carry. Our study gave insight into the barriers and facilitators of acceptance and use of integrating new health technology into existing health systems and findings suggest that the implementation of the UPDX with CAD is feasible in Nigeria with very good perception on acceptability and ease-of-use for community TB screening activities.

## Introduction

Globally, 10.6 million people fell ill with Tuberculosis (TB) in 2022, resulting in an estimated 1.6 million deaths. This makes TB the 2nd leading cause of infectious disease death worldwide after COVID-19. As a result of the COVID-19 pandemic, there was a 5.6% increase in the number of people who died from TB and an 18% decline globally in the number of TB cases diagnosed and notified by the National TB programs, from 7.1 million in 2019 to 5.8 million in 2020, with a partial recovery to 6.4 million in 2021 [1].

In Nigeria, although the COVID-19 pandemic affected efficient TB program implementation, the country was among 4 countries worldwide that conversely experienced an increase in TB case finding within the period of the COVID-19 pandemic [1]. The country however still accounts for 4.5% of the Global TB burden and is the 4th highest contributor to Global missing TB cases [1].

The World Health Organization (WHO) advocates actions to bridge the gap in missing TB cases by scaling-up the deployment of new tools for TB diagnosis and treatment and focuses on intensified research and rapid uptake of new tools and innovation [2]. One of such tools recommended for use is the ultra-portable digital CXR (UPDX)with Computer-Aided-Detection (CAD) [3]. Chest radiography (CXR) plays an important role in the detection of pulmonary tuberculosis (TB) and studies have shown that a large proportion of persons with active TB do not have classical TB symptoms and lung abnormalities due to TB can be detected early with the help of CXR [4]. In many hard-to-reach settings, challenges associated with costs, infrastructural requirements, human reader availability, inter-reader variability and low specificity are known to affect utility of CXR [5]. In these settings, digital CXR systems mounted on mobile units/truck or ultra-portable systems that can be easily carried as a backpack or suitcase can be an important catalyst for greater access to, and equity in high-quality TB care. A number of UPDX systems have been developed and are currently in use with two brands listed on the Stop TB Partnership's Global Drug Facility – Delft Light Backpack and the Fuji Air. The UPDX in use is the Delft Light, Delft Imaging Systems, the Netherlands and CAD software is CAD4TB, version 7.The Delft Light UPDX comprises X-ray generator TR 90/20 (manufactured by Mikasa), X-ray detector CXDI 702-C with accompanying application software (Canon NE) and HP laptop.

Although the current WHO guidelines recommend TB diagnosis on the basis of bacteriological or molecular findings, it has also recently recommended CXR with the CAD Artificial Intelligence (AI), for use in screening and triaging algorithms to assist

with early detection of TB in adults >15 years of age [3]. Children and adolescents under 15 years who are close contacts of TB cases should undergo TB screening using a symptom screen (any one of cough for more than 2 weeks, fever for more than 2 weeks or poor weight gain in past 3 months) or CXR, or both [6]. Prior screening with CXR can determine the effective allocation of molecular WHO-recommended rapid diagnostic tests (mWRDs) to improve case detection and cost-efficiency [7] and again, early TB detection and treatment initiation while using CXR also reduces the likelihood of onward transmission [8]. In Nigeria, the National TB and Leprosy Control Program (NTBLCP) has adapted the WHO guidelines for use of UPDX for early diagnosis of TB in children 4 years and above and adults. The CAD software uses AI to analyze the CXR image for abnormalities suggestive of pulmonary TB, it then automatically interprets this to generate a numerical abnormality score which indicates the likelihood of TB. This abnormality score is used to determine the need for a follow-on mWRDs for TB based on a selected threshold. Studies have shown that the CAD software performed as well and even better than human readers in interpreting plain CXR when screening for pulmonary TB. [9] This attribute can help address challenges around inter-reader variability and reduce delays in reading radiographs when skilled personnel are scarce.

Studies assessing the safety and performance of the UPDX with CAD have shown that these systems operate within the manufacturer's reported emission parameters and leakage doses, which are well below the international guidelines on exposure [7,10,11]. Performance results show that the UPDX systems with CAD used during community based ACF campaigns detect significantly higher CXR abnormality rates and TB case yield [10]. The system's lightweight setup and portable mobile nature provides leverage for reaching vulnerable persons with limited access to medical care in their communities. It has also been shown that extending TB care closer to the homes of people affected by TB can result in greater case detection and the provision of more people-centered care [12,13]. Current WHO recommendations excluded children less than 15 years from the use of CAD systems despite having a number of CAD products in the market, including CAD4TB [14]. As a result of this, there is paucity of data on CAD use and performance scores in children with most studies on CAD use being conducted in populations 15 years and above [10,15–17] and only recently has the first study on CAD use in children 4–14 years in Nigeria been published [18]. There is an urgent need to better document the algorithm performance among different sub-populations including children [16,19] and hence in Nigeria, the National TB and Leprosy Control Program (NTBLCP) has adapted the endorsement of use of UPDX for early diagnosis of TB in children 4 years and above and adults [18].

As more country TB programs adopt and scale-up the use of UPDX systems for TB ACF, it is important to consider the program implications. Although there have been a number of publications describing the experience and results from the use of digital X-rays and CAD in several countries, none has specifically focused on the factors around feasibility and ease-of-use in scale up. A systematic review of factors influencing the scale-up of public health interventions in low- and middle-income countries (LMICs) reported that the most prominent factors influencing scale-up was the availability of financial, human and material resources and most importantly availability of a strategic plan for scale-up including training and supervision. The policy environment, including lack of understanding of CAD software–particularly threshold score selection, lack of international and national guidance on radiation protection suitable for UPDX, community and stakeholder partnerships/collaborations, including knowledge/coordination among stakeholders and having TB diagnoses accepted by health system, availability of data and the efficiency of the supply chain management were also identified as factors that could facilitate or act as barriers to scale-up. [20,21]

Advocacy activities positively influenced scale-up, and the most outstanding were the availability of a strategic plan for scale-up, targeted provider training to ensure that people diagnosed with tuberculosis by new modalities receive prompt treatment and the way in which training and supervision was conducted. Furthermore, collaborations such as community participation and partnerships facilitated scale-up, as well as the availability of research and monitoring and evaluation data [20–23]

A number of studies which assessed the feasibility of scale up of new tests for diseases of public health importance adapted a conceptual framework to explore the feasibility of new health technology introduction [22,23]. Under the

framework, the concept of feasibility is broadly categorized into two inter-related domains, acceptance and usability which are further divided into six sub-domains: learnability, willingness, suitability, satisfaction, efficacy and effectiveness, attributes which are described in contextual detail in the methodology. (Fig 1) [22,23]. According to the authors, the framework recognizes that acceptance and usability may be influenced by factors related to the end-user (both Health Care Workers (HCW) and client), the diagnostic tool and the health system. Health system factors include guidelines and training, quality monitoring and evaluation, supply chain, and policy, budget and planning. All of these factors may play a critical role in the feasibility of scale-up of the use of UPDX with CAD in resource-constrained countries.

In 2021, Nigeria received 10 UPDX (Delft Light, Delft Imaging Systems, the Netherlands) with CAD (CAD4TB, Delft Imaging, the Netherlands) from the STOP TB Partnership, as part of the USAID-funded introducing New Tools Project (iNTP). The project aimed to provide access to high quality, innovative screening and diagnostic tools for TB among populations in hard-to-reach areas. This study seeks to explore the operational feasibility of the scale up of the use of the UPDX with CAD in Nigeria and aims to explore the perspectives of health workers and clients on the operational feasibility and ease of use of the UPDX with CAD in Nigeria, with a view to assessing the acceptability, enablers and barriers to effective use as well as exploring the perceived ease-of-use experience (user experience) of end-users of the UPDX and CAD in Nigeria.

## Materials and methods

### Study design

A cross-sectional descriptive study, conducted between 6th of April 2023 and 15th of June 2023, using mixed quantitative and qualitative methods to determine the feasibility, acceptability, and ease-of-use of the ultra portable digital X-ray with CAD in Nigeria. These attributes work in an interrelated way to contribute to acceptance and use of a new technology. Acceptance comprises positive perceptions, beliefs, and attitudes toward UPDX with CAD and test results among users, i.e., health workers and clients. Use refers to the actions taken by health workers to apply the tool and its results to achieve specified outcomes. If acceptance and use are high, then implementation is feasible. This conceptual understanding informed more focused inquiries in the design of the data collection tools.

The exploratory nature of the objectives gave rise to the qualitative aspect which employed semi-structured interviews with a range of respondents, including key informants, radiologists, radiographers, other healthcare workers (HCWs) and clients. This range of participants ensured that the views of decision makers, end-users of the equipment, and beneficiaries, all of whom influence the overall feasibility and acceptability of roll-out, were explored.

### Study context/setting

This study was conducted in 8 of the USAID-funded TB LON project states implementing the UPDX with CAD intervention which included Kano, Katsina, Nasarawa and Benue in the North while Cross River, Delta, Osun and Oyo States are located in the southern part of the country (Fig 2). Osun and Oyo are part of the TB-LON 3 project implemented by IHV

**Fig 1. Conceptual framework for exploring acceptance and use of UPDX with CAD when introduced into the TB program for Active case finding among targeted communities and Most-At-Risk-populations.**

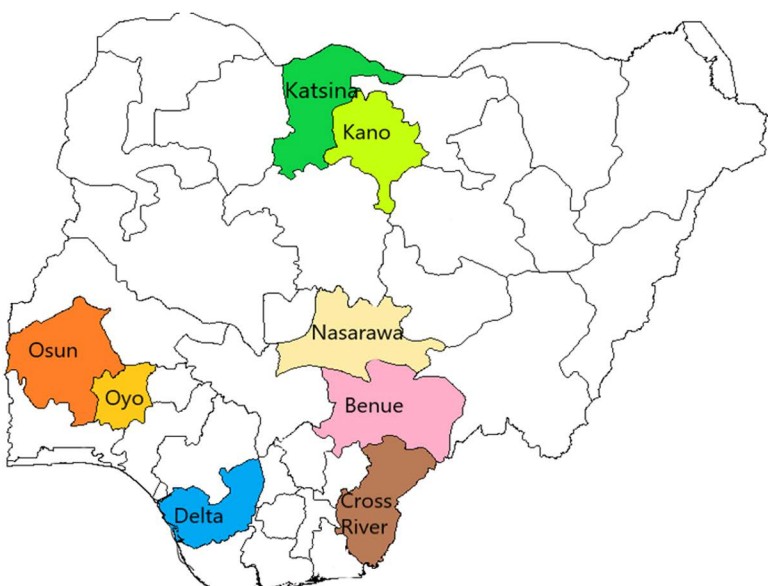

**Fig 2. Map of Nigeria indicating States implementing the UPDX under the iNTP** https://gadm.org/maps/NGA.html.

Nigeria, while the remaining 6 states are under TB-LON Regions 1 & 2, implemented by KNCV Nigeria. The UPDX with CAD were used by the USAID-funded TB LON projects to screen for TB during community active case finding activities in various TB hotspots, key populations, and hard-to-reach areas. A parallel screening algorithm was used in order to identify TB in children and adults 4 years and above, who even though they live in TB hotspots, would not otherwise have accessed health care for TB.

## Study population

The study is part of the Stop TB Partnership's introducing New Tools Project funded by USAID and implemented by KNCV Tuberculosis Foundation Nigeria and IHV Nigeria through the TB-LON Regions 1, 2 & 3 projects. Participants were identified among Radiographers, Program managers/Implementing Partners staff, Radiologists, Tuberculosis and Leprosy supervisors (TBLS)/DOT Nurses and clients known to have used UPDX with CAD for TB screening and all respondents were drawn from the 8 study states, across all 10 teams. Basic training on the use of the UPDX was conducted for the different cadres of participants prior to implementation.

Key informants were senior technical/implementing partners and program managers, and likely to have a broad, strategic overview of issues which should be addressed for successful intervention, and hence their inclusion into the study. Radiologists provided a perspective on the acceptability and ease of use of the CAD readings and image quality of UPDX. Radiographers and other healthcare workers provided first-hand perspective on operating the equipment and its ease-of-use, as well as acceptability. Lastly, clients provided responses on how acceptable it was for them to be screened with these UPDX. All these perspectives were considered to achieve the aim of the study.

## Sampling approach/sample size

The sampling approach for the qualitative component was non-probability sampling, where purposive sampling was used to select the respondents best placed to give information-rich and relevant accounts to fulfill the study's objectives. The study team also engaged research assistants who were also HCW working in the field with the respective respondents

and went ahead to counsel and build rapport with the other participants, putting them at ease while explaining details of the research for them to give honest and accurate responses. The larger qualitative component included 4 key informants, 7 radiologists, 10 radiographers, 16 HCWs and 20 UPDX-screened clients. This brought the total sample size to 57, which created an equal mix between sites and ensured even distribution of the study population. Consenting eligible HCWs (to include LGTBLS/DOTS staff) and consecutive consenting clients for each UPDX team were recruited into the study.

## Data collection

This qualitative evaluation used semi-structured interviews with healthcare workers, key informants, and clients. The radiographers' questionnaire was a mixed methods study which was primarily qualitative, but with some quantitative data collected from the 10 radiographers operating the 10 UPDX. Likert scale questions had scores of between 1 and 5 assigned to the radiographers, with a score of 1 representing strongly disagree, and score of 5 for strongly agreed.

**Study tools: Questionnaire and interview topic guides.** Topic guides were used to conduct semi-structured interview utilizing closed and opened ended questions, with probe questions for further clarifications. The questionnaire domains assessed technical and operational aspects of the use of the UPDX with CAD by the operators such as initiation of UPDX screening, technical operations, Power, maintenance and safety, training and support, knowledge and perceptions, willingness to be screened, willingness to wait for results, perception of tests, challenges/benefits experienced using the UPDX etc. The tailored topic guides were also developed to guide the semi-structured interview with the different participants using the conceptual framework for exploring feasibility and acceptance of new technology introduction (Fig 1). As illustrated in Fig 1, it can be presumed that UPDX acceptance and use are potentially influenced by attributes related to users, i.e., health workers and clients/patients, as well as the diagnostic tool and the health system. These attributes, including learnability, willingness, suitability, satisfaction, efficacy, and effectiveness have been identified in other settings [22,23]. For this study these attributes were adapted with the meanings below;

1. Learnability: ability of the health worker to understand how to correctly perform TB screening using the UPDX with CAD, a new health technology, and accurately interpret the test results.

2. Willingness: Health care workers' intention to carry out screening for TB using the UPDX with CAD among target population, conduct further bacteriological evaluation, and prescribe medication (or not) in line with national guidelines and test results. Regarding the clients, willingness was defined as clients' intention to have the test performed on themselves or their child, wait for test results, and take medication (or not) in line with the test results.

3. Suitability: Health care workers' belief that the screening test using the UPDX with CAD is relevant for his/her work and that test results are a true indication of the suspicion of TB among screened clients and would require further evaluation via mWRDs. Regarding the clients, suitability was defined as clients' belief that the test is relevant in determining whether or not they are presumptive for TB and require further evaluation.

4. Satisfaction: A HCW's feeling that the test is convenient to perform and that it is a process he/she likes doing. It refers to the ease-of-use of the UPDX, which is affected by the design of the UPDX and its instructions for use. Regarding the Clients, satisfaction is described as a feeling that the UPDX test is convenient to take and that it is a process they would like to carry out again or recommend to family and friends.

5. Efficacy: This implies that the HCW is able to make the effort and time to perform a test, read, interpret, and record test results, as well as prescribe the next steps for evaluation, follow-up and treatment where necessary, in line with the test results, as part of the routine TB service delivery.

6. Effectiveness: This refers to the enabling organizational and supporting systems, such as training, supervision, job aids, commodities and supplies, transport logistics, space, and disposal being present or carried out and are integrated into existing routine systems.

**Interview process.** The respondents were approached via email with an attached information sheet/consent. The questionnaire for radiographers was self-administered for the quantitative and qualitative components to elicit appropriate responses. The clients had exit interviews at sites following screening with UPDX, after due explanation of the study objectives and obtaining consent. All interviews were conducted by research assistants in English with consenting health workers and clients. The research assistants engaged were project staff assisting the UPDX radiographers. They conducted "warm-up" interviews to put the respondents at ease to be more likely to give honest, open answers [24], following which the actual interview questions were administered. The study team then carefully reviewed the responses to ensure that all questions received a response and asked any follow-up questions. Interviews were audio-recorded, with field notes and researcher's observations incorporated, to add further depth to the qualitative dataset. The key informants from the IPs were interviewed by one of the researchers who was not a staff of the same organization as the key informant.

**Quality control.** All questionnaires were pre-tested. The research assistants were trained on how to use the tools and were regularly supervised at various levels throughout the data collection period, with two of the supervisors being study co-investigators. All completed data collection tools were checked for accuracy and completeness at the end of each collection through phone calls to respondents such that incomplete or unanswered responses were represented for completion, which allowed for questions to be refined [25]

### Data handling/analysis

Data were first separated into quantitative and qualitative forms prior to analysis. Every sociodemographic information, the radiographer's 5-point Likert scale responses, and other quantitative responses were analyzed using Microsoft excel and IBM SPSS version 26. The study team determined the method of analysis and presentation of the data. Descriptive analysis was done, with frequencies and proportions described in tables and charts [26]. The qualitative dataset was analysed using Taguette version 1.41/1.42. Audio recordings were transcribed verbatim after the interviews and the transcripts served as the data for analysis, which was later uploaded into the Taguette app. Deductive coding was conducted into themes based on ideas, experiences, feelings, opinions, perceptions and beliefs of respondents [27,28] and were subsequently organized and presented into a conceptual framework for exploring acceptance and ease of use/usability (Fig 1) [22,23]. The themes generated were further organized into facilitators and barriers and disaggregated among the different groups of HCW respondents and the clients Fig 3 [22]. Parametric tests can be used to analyze Likert scale responses but to describe the data, means are often of limited value unless the data followed a classic normal distribution where frequency distribution of responses was done as in the study [26].

### Ethical approval

Ethical clearance was obtained from the National Health Research Ethics Committee with approval number NHREC/01/01/2007-03/04/2023 and the study conducted according to the principles of World Medical Association Declaration of Helsinki. Written and signed/thumb printed informed consents were obtained from the study participants and caregivers where applicable.

### Results

The perspectives of 57 respondents from 10 UPDX sites in 8 states, which aim to assess the feasibility and acceptability of introducing UPDX with CAD for use in Nigeria were evaluated. The participants were healthcare workers (including key informants) and clients and they included radiographers 10 (17.5%), radiologists 7 (12.3%), key informants (senior technical/implementing partners and program managers), 4 (7%), TBLS/DOT nurses 16 (28.1%), and clients 20 (35.1%) whose occupations are shown in Table 1. Among the respondents, 43 (75.4%) were males, 14% (8) were 20–30years old, 36.8% (21) were aged 31–40 years, 28.1% (16) were 41–50 years, while the remaining 21.1% (12) were 51–60 years old.

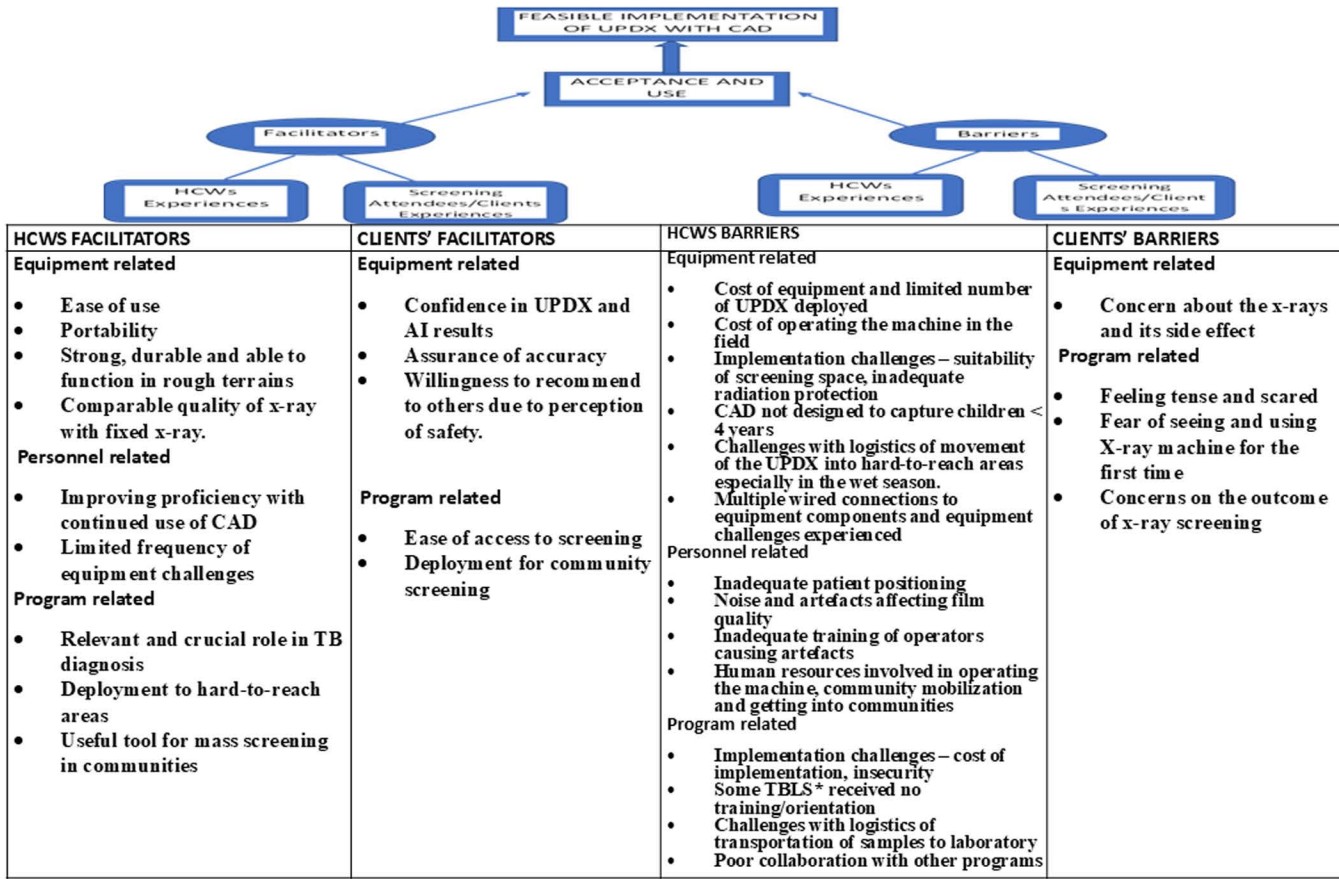

**Fig 3. Illustration of key facilitators and barriers of the acceptance and use of UPDX with CAD in the early implementation phases in Nigeria (Adapted from Ansbro ÉM 2015).**

In general, 8% of the HCWs have worked in the TB program for more than 20 years, while 34/37(92%) have worked for 20 and less years. Eighty percent (80%) of the Radiographers have been in the TB program for less than 4 years while 20% have worked for 5–9 years. Forty percent (40%) of them have previous experience with UPDX with CAD while 60% have none. Other socio demographic characteristics of respondents are shown in (Table 1).

## Results of the radiographer's perspectives

The Radiographer's questionnaire was structured into 5 inter-related domains with mixed method responses for acceptability, feasibility, ease/speed of use as depicted in the 5-point Likert scale (Table 2). They reported ease of use, portability, availability in the community, rapid results and had challenges with equipment/implementation costs, fears of irradiation and lack of collaboration with other programs. All (100%) strongly agreed that the UPDX/accessories were easy to set up, considered themselves proficient with operating the UPDX and to a lesser extent agreed that the programs and software are user-friendly and easy to set up. They however disagreed that the device is portable enough for one person to carry and had varied opinions on the quality of images, adequacy of the power source in providing power throughout the day, ease of transfer of images to other devices and comparison of the image quality of the UPDX with normal CXRay (Table 2 and summarized in Table 3).

**Table 1. Socio-demographic characteristics of the 57 respondents.**

| Variable | Frequency (%) |
|---|---|
| **Highest Educational Qualification of 37 Health Workers** | |
| None | 0 (0) |
| Primary/First School Leaving Certificate (FSLC) | 0 (0) |
| High School/West African School Certificate (WASC) | 0 (0) |
| College (DIPLOMA) | 6 (10.5) |
| University (BSC_BA) | 20 (35.1) |
| MSc | 3 (5.3) |
| MD/MBBCH | 1 (1.75) |
| PhD/FELLOWSHIP | 7 (12.3) |
| **Highest Educational Qualification of 20 Clients** | |
| None | 1 (1.75) |
| Primary/First School Leaving Certificate (FSLC) | 1 (1.75) |
| High School/West African School Certificate (WASC) | 11 (19.3) |
| College (DIPLOMA) | 5 (8.8) |
| University (BSC_BA) | 2 (3.5) |
| MSc | 0 (0) |
| MD/MBBCH | 0 (0) |
| PhD/FELLOWSHIP | 0 (0) |
| **Work experience of 37 Health workers in TB program (in years)** | |
| 1–10 | 17 (46) |
| 11–20 | 17 (46) |
| 21–30 | 3 (8) |
| **Occupation of 20 clients** | |
| CHEW | 1 (5) |
| Driver | 1 (5) |
| Business | 5 (25) |
| Trader | 4 (20) |
| Farmer | 4 (20) |
| Civil servant | 1 (5) |
| Corper/student | 3 (15) |
| Undisclosed | 1 (5) |

**Initiation of UPDX screening.** All (100%) of the radiographers strongly agreed that they themselves were proficient with operating the UPDX system machine and that the UPDX machine and accessories/components are quite easy to assemble and set-up. One respondent said "*Because of my consistent use of the machine I have built myself and understood the mode of operation of the machine. Also, due to my keen interest during any troubleshooting, I can manipulate the machine if any challenge arises*" and another gave a reason as" *I have navigated the system for almost 2 years and have gotten acquainted with it*".

A significant proportion (60%) agreed that the programs and software on the systems are user-friendly and easy to operate. On the average, it took 10 minutes, with a mean of 10 ± 1.3, for the UPDX with CAD system to be set up for screening at a location, with at least 2 persons needed to support commencement of screening activity at a site.

PLOS Global Public Health

**Table 2. Likert Scale responses on acceptance and ease of use of the ultraportable digital CXR with CAD by radiographers in Nigeria (N=10).**

| VARIABLES ON THE 5-POINT LIKERT'S SCALE | FREQ | Strongly Disagree | Disagree | Neutral | Agree | Strongly Agree | Mean ±SD |
|---|---|---|---|---|---|---|---|
| The UPDX machine and accessories/components are quite easy to assemble and set-up? | 10 | 0 | 0 | 0 | 0 | 10 | 5.0±0.000 |
| Programs and software on the systems are user-friendly and easy to operate? | 10 | 0 | 0 | 0 | 6 | 4 | 4.4±0.516 |
| Consider self-proficient with operating the UPDX system machine? | 10 | 0 | 0 | 0 | 0 | 10 | 5.0±0.000 |
| The image quality of the UPDX is not different from a normal CXR quality? | 10 | 1 | 0 | 3 | 3 | 3 | 3.7±1.252 |
| Transfer of CXR images from the UPDX system to another device is easy? | 10 | 0 | 0 | 1 | 4 | 5 | 4.4±0.699. |
| Adapting the device settings for the UPDX Generator to the different sizes of clients screened is easy? | 10 | 0 | 1 | 1 | 7 | 1 | 3.8±0.789 |
| A fully charged Generator is adequate for screening clients through-out working the day | 10 | 0 | 3 | 1 | 4 | 2 | 3.5±1.179 |
| I often experience technical challenges with the UPDX system | 10 | 0 | 7 | 0 | 3 | 0 | 2.6±0.966 |
| Do you think the device is portable enough for 1 person to carry? | 10 | 9 | 1 | 0 | 0 | 0 | 1.1±0.316 |
| The UPDX power source is adequate for providing power through-out the day for screening clients? | 10 | 0 | 0 | 4 | 2 | 4 | 4.0±0.943 |
| Conducting daily maintenance on the machine is easy | 10 | 0 | 5 | 2 | 2 | 1 | 2.9±1.101 |
| The radiation safety profile of the UPDX is satisfactory? | 10 | 1 | 1 | 1 | 6 | 1 | 3.5±1.179 |
| The radiation safety measures provided for your protection adequate? | 10 | 2 | 4 | 1 | 3 | 0 | 2.5±1.179 |
| Training for the UPDX operation was adequate and sufficient for daily operations with the machine. | 10 | 0 | 0 | 2 | 3 | 5 | 4.3±0.823 |
| Quality of post-training technical support (remote and physical) is adequate? | 10 | 0 | 1 | 0 | 5 | 4 | 4.2±0.919 |
| Easy reaching the DELFT support staff and getting the required assistance when in need of support on the field? | 10 | 0 | 0 | 0 | 1 | 9 | 4.9±0.316 |

The average number of persons that could be screened daily using the UPDX was a mean of 93±1.4 with an average mean of 6.1±0.9 hours/day spent out in the field screening clients. Also, on an average, it took a mean of 14±1.1 days for the radiographers to get comfortable/proficient with operating the machine.

**Technical operations.** Almost all (90%) radiographers agreed that transfer of CXR images from the UPDX system to another device was easy and 60% of them stated that the image quality of the UPDX is not different from a normal CXR quality. On the average, it takes a mean of 60±1.9 seconds to transfer images wirelessly from the detector to the laptop for CAD processing. 70% of them agreed that adapting the device settings for the UPDX generator to the different sizes of clients screened was easy. In response to how easy it was to adapt the device settings for such, one radiographer responded *"this is not wholly easy depending on the population you are screening, you have to adjust and re-adjust"* and another said *"it delays the screening"*.

On another note, 70% disagreed that they often experience technical challenges with the UPDX system, and one submitted *"I had challenges at the early stage of using the UPDX but ever since then have not encountered any challenges again"*. Majority stated the frequency of experiencing technical difficulties as *"once in a while"*, *"once in a blue moon"*, *"once in 2-3 months"*, and one expressed *"it could go on for months without any challenge except if a component*

Table 3. Quantitative responses on useability of the ultraportable digital CXR with CAD by radiographers in Nigeria (N = 10).

| Variables with string responses | FREQUENCY (%) | Mean±SD | Minimum (%) | Maximum (%) |
|---|---|---|---|---|
| Average time to set up UPDX system for screening? (mins) | 10 (100) | 10 ± 1.3 | 3(10.0) | 15(10.0) |
| Number of persons needed to support commencement of screening activity? | 10(100) | 1.5 ± 1.2 | 0(30.0) | 3(20.0) |
| On the average, how many persons can you screen daily using the UPDX? | 10(100) | 93 ± 1.4 | 60(10.0) | 200(30) |
| On the average how many hours/days do you spend out in the field screening clients? | 10(100) | 6.1 ± 0.9 | 4(30.0) | 8(40.0) |
| How many days or weeks of operations it took to get comfortable/ proficient with operating the machine? (in days) | 10(100) | 14 ± 1.1 | 3(10.0) | 60(10.0) |
| How long (secs) to transfer images wirelessly to the system? | 10(100) | 60 ± 1.955 | 10(10.0) | 300 (10.0) |
| How long does your battery power last when you are on the field? (hours) | 10(100) | 6.2 ± 1.814 | 4(30.0) | 24(40.0) |
| How many exposures on the average can you capture on a fully charged battery? | 10(100) | 92 ± 1.897 | 60(10.0) | 200(10.0) |

*becomes bad which does not occur often. Sometimes the awareness of the operator matters too."* However, when challenges do occur, the commonest technical difficulties experienced while using this machine were *"connectivity problems where the detector connection to the CAD software on the laptop don't come up as well as the wired LAN connection between CAD4TB box and wireless router not coming up".*

Ninety (90%) strongly disagreed that the device is portable enough for 1 person to carry, with majority insisting "*The device requires more than the one person suggested.*" The preferred deployment method for the majority (90%) using CAD/AI technology to interpret chest x-rays was *"hybrid: using locally installed CAD to parse X-ray offline and backup/ synchronization when there's internet connection or on demand."*

**Power.** A fully charged battery/power bank was not always adequate for screening clients through-out the working day, but 100% of the radiographers agreed that recharging the UPDX with the supplementary power source (a Mobisun solar panel) is adequate for providing power through-out the day for screening clients. All (100%) of them usually use Mobisun battery and solar panel as power source on the field while screening and submitted that on the average, the battery power lasts a mean of 6.2 ± 1.8 hours when out on the field and a mean of 92 ± 1.9 exposures can be captured on a fully charged battery.

**Maintenance and safety.** Majority (90%) of the radiographers strongly agreed that it was easy to reach the DELFT support staff and obtain the required assistance when in need of support on the field.

There were however mixed reactions from them on ease of conducting daily maintenance on the machine, satisfactory radiation safety profile of the UPDX as well as adequate radiation safety measures provided for their protection. The respondents raised some concerns about the safety of the UPDX. The commonest was the lack of adequate shielding, especially regarding the lack of a thyroid protector.

According to one respondent, *"The thickness of the lead apron is light and there is no protection safety gadget for the eye, skin and thyroid. Also, there is no lead shield for protection, if I have to expose myself for up to 70 client on a daily basis".* Another respondent added," *The two other radiation safety measures are not observed which is Time and shielding. Time: the Less time spent near radiation source less radiation received. Shielding behind shield from radiation source the less radiation received. For both the Radiographer and the Data Clerk. Shielding will help other parts of the body without lead apron because the eyes and thyroid are radiosensitive. Survey meters are not provided to establish radiation field/zone which has to do with safe distance and for research purposes."* The other safety concern was the duration of staff exposure which a respondent suggested should be minimized. Other suggestions were" *Having a local maintenance*

team in-country should be another part to be looked into and also having replacements components in country", "Periodic checks on the x-ray tube head for radiation leakage" and "Provision of Lead shield or other radiation PPE's like googles, thyroid shield. Provision of immediate readable TLD badge".

**Training and support.**  All (100%) of the radiographers agreed that the training they received for the UPDX operation was adequate and sufficient to enable them successfully carry out daily operations with the machine, with the majority (90%) submitting that the quality of post-training technical support (remote and physical) received was adequate. They suggested recommendations for improvement on the UPDX system to make their work easier and these included *"Bulkiness of the system, if it's possible to implement a total wireless connection system for it to be easy to carry about", "PDX system should be upgraded" with a" Reduction in the number of cables for the set up if possible" and" A remote adjustable detector on a pole instead of the manually adjustable cable".* One respondent mentioned support for *"radiographer's welfare"*

## Results of analysis of qualitative responses from other participants

**General comments.**  There was disagreement by the Radiologists on the device being portable enough to be carried by one person and a summary of the views of various respondents on the portability/bulkiness of the UPDX with some qualitative features as follows:

*"The portability of the UPDX has made it enter remote areas giving ease of accessibility because the machine can be taken to places difficult to access but when we went to a distant community without mobile network, the machine developed connectivity issues"*

Concerning the bulkiness of the system, respondents felt *"there should be reduction in the number of cables for the set up or if it's possible to implement a total wireless connection system for it to be easy to carry about". "One respondent suggested a software application exposure knob on the laptop where you can click on an App and the patient is exposed".*
*Again, that the UPDX system should be upgraded with further work on the AI as it misses microcalcifications, and less often assumes all densities are related to pathology which is not so". "The manufacturer needs to improve on differentiating pathologies from physiological findings"*

*"I believe AI is a technology that requires careful management and the manufacturers should be able to communicate how the technology works in a way that others can understand, and also Identify the ethical implications of using AI in clinical contexts". "The down side is AI(CAD) lacks the capacity to handle ethical and societal challenges with patient care and can dehumanize healthcare and practice"*

*"The only challenges we have had is the movement of the UPDX machine to hard-to-reach areas in my LGA, worse during the rainy season"*

**Clients.**  Majority (100%) of the clients had a good knowledge of tuberculosis and how it spreads through coughing. They said,*" Tuberculosis is a communicable disease that affect your lungs and can be transferred from one person to another. Yes, its possible someone can have it without knowing, because you mustn't be coughing."* On further probing some of them responded,*" Yes it is possible because I only came to see what is happening and I was told I have TB"* and another added,*" Yes me too I don't know I have it I also thought it was just cold and I was treating because it was during harmattan".* One summarized her knowledge of TB as*" Tuberculosis is a dangerous bacteria disease that can be majorly present in a person that coughs more than 2weeks or more. There is no way you can know your TB status unless you get tested for TB. It is possible for some individuals who are infected with TB to not exhibit any symptoms, making it difficult for them to know their infection status unless they undergo TB Testing and willingness to be screened with UPDX_CAD"*

Many of the clients had confidence in UPDX and AI results and reported several advantages of the UPDX such as ease of access, route of access, faster results due to the use of CAD, mobility and portability. The clients liked the fact that the UPDX machine came to meet them in the communities. In their words,*" it was very easy for me and timings were convenient as it was in my community where I don't have to travel to another community for the screening"*.

On the route of access, one client said," *I saw them round, when I was passing by the road so I decided to get screened since I was having chest pain as well"* and yet another said,*"I visited a friend working with the Covid-19 team that was with the UPDX team in a joint community outreach and I was encouraged by the friend to get screened, and that was how I was diagnosed"*.

Concerning the cost implications, they were happy they didn't have to pay for the screening and one opined that,*" I am very happy for such opportunity because its free, and accessible at comfort of my house."* Again, it was easy to go for the TB screening because it was free and, in their words," *"We were told to come for free chest x-ray screening for tuberculosis test, by the town crier."* And another said," *It was very easy for me without me paying money for the screening"*.

Majority of the clients had no concerns about the screening with mixed reactions on preference for other testing types but some others were scared about the outcome stating," *At first, I felt really scared and didn't even want to do the X-ray for fear of the outcome. I know TB is a deadly disease because one of our friend was diagnosed and seeing my picture with that color and I was told it an infection I was more scared"*, and yet another said," *I felt tense and scared of becoming a patient. It is my first time of seeing X-ray machine live and first time of doing it"*. Some thought it would be painful saying," *I felt scared at first because I was thinking it will be painful"* and one showed concern about the rays and their side effects, *"I have concern about it because of the rays, because of the side effect of the X-ray."*

The clients who were initially scared later relaxed as the research assistants conducted "warm-up" interviews to put them at ease, following which the actual interview questions were administered. All the clients were willing to wait for the final results after sputum testing at the lab, and willing to recommend the screening to others as they benefitted because its free, fast, stress free, easy to access, time saving and they will get to know their TB status.

**Radiologists.** The radiologists reported numerous advantages of the UPDX such as ability to reach a large number of people from any location, ease of access, route of access, better x-ray resolution than the standard stationary CXR machines, time saving, faster results, high turnover rates, mobility and portability. In response to advantages of the UPDX, one radiologist said that*" The UPDX has an advantage of flexibility and mobility in hard-to-reach areas"* and another reported that *"The portability and digital processing capabilities of the UPDX gives it an edge over the stationary/ conventional x-ray machines. Patients are also met in their locations instead of waiting for them to come to health centers/ clinics."*

Majority of them have not changed their impression of TB burden following reporting these films. On further prompting, one said" Yes *it did. I previously thought TB prevalence had reduced drastically or even eradicated but my engagement with the TB control program has totally reversed this perspective as I now know TB is still here with us."* And other opined that,*" Not really, though I feel the prevalence should reduce if the populace take more active measures"*

The challenges noted included lots of artefacts, motional blurring, lots of noise, and reduced quality of images for large sized people, all of which may be related to the skill of the radiographer.

*"The quality of films from the UPDX is very poor and sometimes the quality is so poor that he can't report such films and has to return the films*

*The quality of the images is good for the portable machine when regular sized people are x-rayed but cannot be compared with images of bigger people when done with the stationary X-rays in the facilities"*

Other challenges highlighted include inability to carry out repeat studies and carry out other views. One said," *The quality of the films from UPDX is better if the stationary machine is using manual processors, however if the stationary*

machine is using digital processors, they are comparable”. Another opined, “The *quality of the images are good for the portable machine when regular sized people are x-rayed but cannot be compared with images of bigger people when done with the stationary X-rays in the facilities.”* Even though the radiographers were trained by the manufacturers, all the radiologists had advices/recommendations for the manufacturers and in their words,” *The manufacturers should be involved in the training of the radiographers in the use of this machines (UPDX) especially with regards to optimal patient positioning and radiation dosage”,” Upgrading strength of the UPDX or having improved software that improve the quality of poor images without compromising the real image quality for radiologists engaged for clinical review of X-rays from UPDX.”*

The Radiologists advised caution on use of AI. One opined, *“I believe AI is a technology that requires careful management; the manufacturers should be able to communicate how the technology works in a way that others can understand, also identify the ethical implications of using AI in clinical contexts.”* Another suggested, *“the down side is AI(CAD) lacks the capacity to handle ethical and societal challenges with patient care and will dehumanize healthcare and practice”* Two respondents suggested” *The manufacturer needs to improve on differentiating pathologies from physiological findings”* and finally,” *To continue to work with radiologists and radiographers in designing all the components of the systems.”*

**Program managers.** All the PMs/IPs were aware of the WHO recommendation on use of CAD and acknowledged the benefits of the roll out of the UPDX with CAD in increased TB case detection, notification, speed of diagnosis, treatment and reduced catastrophic cost. In addition, they highlighted the benefits of increased access and increased workforce efficiency as the CAD will enable non specialist clinicians to make triage decisions. They recognized the tool as an innovative adjunct for TB case search in communities and opined it should be made a compulsory package for all teams on active case search in communities. The drawbacks they reported included,” *cost of acquiring the machine”, “connectivity challenges and faulty parts that will require the intervention of the manufacturers to send replacement from overseas.”,” Not designed to capture children that are under-5”, “Identifying and assigning high CAD score to other remarkable lesions in the lung as presumptive TB.”,* and” *the number deployed to the community are too few”*

**TB local government supervisors/DOTS nurses.** There were various benefits to the nurses/TBLS which included as stated *“conducting community outreaches have become more easier, Facility screening staff have been more affected (engaged)”* and again” *The benefits of HCW's from this machine will be that community members are more interested in been screened for TB, rather than the manual way of screening using WHO symptom screening.”* One major challenge as described by a TBLS was” *The challenges we have been having is finding patients when they are been screened and diagnosed in a larger community, how to trace and enroll them, especially when they don't have a mobile.”* Others mentioned are *“lack of use of SOP/guidelines and no training/orientation given, among others”* and yet another said *“There is a particular challenge we are encountering and that is all about the machine which is unable to screen children below the age of 5 years”.*

The themes generated from the qualitative analysis were further organized and presented as facilitators and barriers within the different levels of the framework and disaggregated among the 37 HCWs and 20 clients. They reported facilitators being ease-of-use and access to screening, portability, availability in hard-to-reach areas, usefulness for mass screening in communities, comparable quality of x-ray with fixed x-ray, rapid results and had challenges with equipment/implementation costs, fears of irradiation, lack of collaboration with other programs and inability to use UPDX-with-CAD on children < 4 years (as summarized and illustrated in Fig 3).

## Discussion

This qualitative study is the first of its kind in Nigeria, documenting the acceptability, operational feasibility, and ease of use of the UPDX and CAD with emphasis on the barriers and enablers/facilitators to effective use of UPDX. It is also one of the first to explore the perspectives of clients screened using the UPDX with CAD and no other research team has studied facilitators and barriers to UPDX with CAD in Africa. Our study findings showed the respondents (clients) expressed

their confidence in the quality of images and accuracy of results from the CAD and acknowledged the impact on increased access to TB screening, while the TBLS/DOT nurses experienced increased work load/case finding and increased TB detection. This is similar to other studies that have reported HCWs immense satisfaction to the extent that the machine is said to "work like magic" [20] and also documented the significant potential of the UPDX to improve case detection [10,29]. Another study comparing various AI algorithms and human readers in high burden areas documented that AI algorithms can be highly accurate and useful for TB detection and outperform human readers [30]. The perception of benefits identified in our study translated to a willingness of the program staff to recommend a scale up of the use of UPDX with CAD, while clients were willing to recommend screening with UPDX with CAD to other persons, overriding some concerns expressed around safety. This contrasts with another study, which identified issues around HCWs lack of confidence in the diagnosis resulting from CAD-enabled mobile Xray system which led to delayed treatment initiation [21]. Anecdotal evidence from the Nigerian TB program suggests that interventions are more likely to succeed when HCWs and clients are sufficiently confident in the outcome of such intervention.

The Radiologists in the study generally had a positive assessment of the quality of the Xray generated and its capacity to demonstrate abnormalities suggestive of TB compared to stationary Xray, however they noted image qualities to be dependent on clients' body sizes and expertise of the users. This implies a need for implementers to conduct regular re-training and quality checks on the radiographers to ensure they adhere to standards for exposure of clients based on body size. Some studies have documented lower image quality [10] with mean image qualities significantly lower than the reference range [10].

Another usability aspect explored by our study was satisfaction with ease-of-use of the UPDX with CAD. Radiographers in the study were generally satisfied with the ease-of use of the UPDX with CAD with positive perceptions on the user-friendliness of the machine and the software components, ease of set-up, short time to proficiency (average of 14 days), and ease of access to service support from manufacturers when in need of assistance. Regarding number of exposures on a fully charged battery, the verdict was good. Our study found an average of 92 exposures on a fully charged battery with the battery lasting for a mean of 6.2 hours, with one respondent reporting as high as 200 exposures on a full charge. Another study reported x-ray capture capacity 58% lower than the expected 100 exposures per charge [10]. Similar studies done elsewhere also identified the equipment portability/battery power, feasibility, ease of use were key considerations from the user's perspectives [20,22]. A key challenge highlighted by radiographers with use were connectivity challenges between components affecting image transfer and operations requiring service support. This is similar to reports from studies in other settings where similar equipment was used [10,20]. Considering the UPDX is used majorly for community ACF in rural settings and sometimes hard-to-reach areas in Nigeria the satisfaction of end-users with the ease-of-use of equipment becomes a key determinant of acceptability and this impacts on the performance and output. It is critical end-users continue to get quality and responsive service support from manufacturers as some studies have noted that ease-of-use and acceptability may change as the machine gets older and use increases [20].

On the issue of portability, the responses were quite relative. While the radiologists and other HCWs considered the UPDX as being portable, the radiographers who are the handlers of the machine report that it's not portable enough for one individual to carry and operate as advertised. The whole system is considered bulky, causing it to require at least two persons to carry and operate effectively during outreaches. Other studies have also documented that the UPDX is not that portable as advertised [20]. Although our study respondents highlighted increased access of the UPDX to the doorsteps of clients which is facilitated by its portability relative to stationary and mobile Xray/trucks, the need to engage 2 persons to effectively operate the machines for TB ACF in the field increases the operational costs for the machine and was identified as a barrier. This alongside the cost of acquiring the machine are likely to be key considerations while planning for the scale up of its use for TB ACF in a resource-limited setting like Nigeria. Other studies done in hard-to-reach settings have also shown challenges associated with costs and efforts are made to reduce these barriers [5].

Our study has several strengths. The wide and varied distribution respondents interviewed across the cascade of implementation including clients, and across the different states and geographical locations gave a wider variation of the perceptions of the respondents and strengthened the quality of information for policy makers and manufacturers as used in other studies [22]. The ability to obtain accurate views from a wide coverage of respondents of health care workers of different cadres and job descriptions as well as clients of different ages, works of life, educational qualification adds quality to the study findings. Some other studies were limited by the lack of this spread and exclusion of clients [20,22,23]

In addition, the engagement approach of research assistants who went ahead to build rapport with the participants, putting them at ease while explaining details of the research for them to give honest and accurate responses strengthened the quality of responses. Some studies have shown that establishing such rapport actively detects and limits social desirability bias which strengthens the quality of data collected [24].

Using a conceptual framework adopted based on earlier models and publications [22,23,31] facilitated the exploration of programme implementation and may prove useful in other settings.

The principles of learnability, willingness, suitability, satisfaction, efficacy and effectiveness by the HCWs and clients, which this study set out to explore were successfully explored. Other studies have documented successful acceptability and feasibility by HCWs and appropriateness of integration of a new testing procedure into the health care system [32]. The effectiveness of the health system in providing an enabling environment is very important [22].

Enablers and barriers were carefully sieved out manually though our study used Taguette statistical package for majority of qualitative data analysis. Software packages are useful in analysis but should not be viewed as an easy way out or indeed short cuts to detailed systematic analysis. High quality analysis still depends on the skill/experience, vision and integrity of the researcher [25,28].

The main limitation of the study is that it is the first to assess feasibility of a new product since the machine is new in the country, and the respondents contact/use was also relatively new.

Since the products are new, early implementers have a reason to use this study which is based on hands-on/site experiences to advocate for machine and service improvement.

Again, our study is based on only one device and software combination. We do not know if this can be extrapolated to another software on the same portable device.

Additionally, as the respondents are concerned about the quality of images on this device, there is need for more evidence to assess the yield and accuracy of these devices with CAD software. Its applicability in the field will be limited if the software cannot yield scores for such low-quality images by the portable CXR devices.

Several recommendations from end-users arose from this study. Further evaluation of the CAD4TB threshold score used in Nigeria of the CAD4TB and heat map of the AI is necessary to be able to identify all possible TB lesions. Also, periodic safety checks for radiation leakage should not be taken lightly as well as other safety concerns. International safety standards should be updated to consider the new UPDX with very low radiation.

The manufacturers should engage radiographers and radiologists when planning design updates to the different components of the equipment. In addition, it is recommended that there is reduction in number of wire connections and accessories, tripod stand weight to reduce the bulkiness of the system and make set up less complex, as well address technical difficulties to ensure seamless connectivity between the components.

There was also a strong recommendation for the scale up of the use of UPDX with CAD which would also require increased investment in other implementation costs like maintenance, staff remuneration, and logistic support for operating in hard-to-reach areas.

Again, there is need for efficient program implementation of UPDX-CAD to ensure optimal sputum transport logistics to the laboratory, and effective collaboration with HIV, Malaria, Nutrition, and other relevant programs to promote a one-stop-shop approach to health service delivery.

Finally, future steps to implement the UPDX with CAD system are recommended. Studies on cost-effectiveness are urgently needed, with more of the impact studies of UPDX with CAD on TB case finding [18] needed in Africa as was done in Peru and also more studies done for other new TB diagnostic technology, GeneXpert [33] and Truenat [34].

## Conclusion

This study explored the perspectives of health workers and clients on the operational feasibility and ease of use of the UPDX with CAD, giving insight into the barriers and facilitators of acceptance and use of integrating new health technology into existing health systems, with a view to determining the learnability, willingness, suitability, satisfaction, efficacy and effectiveness of the UPDX with CAD by the end-users. Study findings suggest that the implementation of the UPDX with CAD is feasible in Nigeria with a very good perception on acceptability and ease-of-use for community active TB screening activities. The main barriers include the costs associated with acquiring the equipment, implementation costs, logistic challenges and associated equipment challenges

## Supporting information

**S1 Text. Quantitative Radiographer's_SPSS analysis.**
(DOCX)

**S2 Text. Annex 1_ Study Instruments for Operational Feasibility.**
(DOCX)

**S3 Text. Annex2_Informed Consent for Operational Feasibility.**
(DOCX)

**S1 Data. Radiologists' codebook.**
(PDF)

**S2 Data. Radiographers' codebook.**
(PDF)

**S3 Data. Nurses HCW codebook.**
(PDF)

**S4 Data. Codebook for clients.**
(PDF)

**S5 Data. Codebook for PMs and IPMs.**
(PDF)

**S6 Data. Qualitative_Taguette analysis of all HCW and Clients.**
(XLSX)

## Acknowledgments

The authors acknowledge the support by the National Tuberculosis and Leprosy control program, State TB control Managers across the USAID TB-LON supported study states in Nigeria, Tuberculosis Local Government supervisors, Health care workers, TB-LON project staff in the selected sites and study participants for their contribution to the success of this study. We also appreciate the technical support and input from the Stop TB Partnership introducing New Tools Project (iNTP) team and USAID Washington TB Division.

## Author contributions

**Conceptualization:** Sani Useni, Rupert Eneogu, Austin Ihesie, Abiola Alege.

**Data curation:** Sani Useni, Rupert Eneogu, Austin Ihesie, Abiola Alege, Atana Ewa.

**Funding acquisition:** Rupert Eneogu, Austin Ihesie.

**Investigation:** Sani Useni, Eze Chukwu, Jamiu Olabamiji, Atana Ewa.

**Methodology:** Sani Useni, Rupert Eneogu, Austin Ihesie, Abiola Alege, Aderonke Agbaje, Bethrand Odume, Debby Nongo, Eze Chukwu, Jamiu Olabamiji, Omosalewa Oyelaran, Charles Ohikhuai, Zhi Zhen Qin, Rachael Barrett, Olufunmilayo Omosebi, Chukwuma Anyaike.

**Project administration:** Sani Useni, Rupert Eneogu, Austin Ihesie, Abiola Alege, Aderonke Agbaje, Bethrand Odume, Debby Nongo, Eze Chukwu, Jamiu Olabamiji, Omosalewa Oyelaran.

**Resources:** Sani Useni, Rupert Eneogu, Abiola Alege.

**Supervision:** Atana Ewa.

**Validation:** Sani Useni, Rupert Eneogu, Austin Ihesie, Abiola Alege, Eze Chukwu, Jamiu Olabamiji, Atana Ewa.

**Visualization:** Sani Useni, Austin Ihesie, Abiola Alege, Atana Ewa.

**Writing – original draft:** Sani Useni, Rupert Eneogu, Austin Ihesie, Abiola Alege, Zhi Zhen Qin, Rachael Barrett, Atana Ewa.

**Writing – review & editing:** Sani Useni, Rupert Eneogu, Austin Ihesie, Abiola Alege, Aderonke Agbaje, Bethrand Odume, Debby Nongo, Eze Chukwu, Jamiu Olabamiji, Omosalewa Oyelaran, Charles Ohikhuai, Zhi Zhen Qin, Rachael Barrett, Olufunmilayo Omosebi, Chukwuma Anyaike, Atana Ewa.

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
