## [Decision Letter · Decision Letter 0]

24 Feb 2025

PGPH-D-24-03076

Operational feasibility of the Ultra-Portable digital X-rays with Computer-Aided Detection (CAD) for community active case finding for TB in Nigeria: Health care workers and client’s perspectives

Dear Dr. EWA,

Thank you for submitting your manuscript to PLOS Global Public Health. After careful consideration, we feel that it has merit but does not fully meet PLOS Global Public Health’s publication criteria as it currently stands. Therefore, we invite you to submit a revised version of the manuscript that addresses the points raised during the review process.

I agree with the reviewers that this manuscript adds valuable contributions to the understanding of implementing digital X-rays with CAD for TB diagnosis. I encourage the authors to pay careful attention to the comments of the reviewers. In particular, please note the following:

- As Reviewer #1 mentioned, Table 2 could use significant improvement in presentation and usefulness of the data presented.

- I also agree with Reviewer #2 that the abstract could be substantially improved by describing key findings in greater detail. In addition, please specify that the UPDX with CAD was used for TB screening, not for diagnosis and treatment per se.

- Please also note Reviewer #3's comment that the specific UPDX and CAD systems should be clearly and explicitly named in the beginning as the acceptability and feasibility profiles are likely to be system specific.

In addition to the Reviewers' comments, my own comments are as follows:

- Please add more details about the TB screening programs in which the UPDX with CAD were used. Some of the text refers to convenience of being tested at home and others mention stopping by at a community event., so the reader gets glimpses of how TB screening was done. However, it would be better to explicitly state the different settings and programs in which TB screening was done.

- Line 568 states that the study reports "unbiased" views. Please consider using a different term or explain how the reported qualitative data can be considered "unbiased".

- Lines 572 - 576 in Discussion mentions the importance of building rapport with participants. However, this is not mentioned in Results and not sufficiently described in the Methods. Please explain in Methods what was done to build rapport and in the Results, the evidence that shows that rapport was successfully established.

We look forward to receiving your revised manuscript.

Kind regards,

Sanghyuk S Shin

Academic Editor

Journal Requirements:

1. Figure 2: please (a) provide a direct link to the base layer of the map (i.e., the country or region border shape) and ensure this is also included in the figure legend; and (b) provide a link to the terms of use / license information for the base layer image or shapefile. We cannot publish proprietary or copyrighted maps (e.g. Google Maps, Mapquest) and the terms of use for your map base layer must be compatible with our CC-BY 4.0 license.

Additional Editor Comments (if provided):

Reviewers' comments:

Reviewer's Responses to Questions

**Comments to the Author**

1. Does this manuscript meet PLOS Global Public Health’s publication criteria?

Reviewer #1: Yes

Reviewer #2: Yes

Reviewer #3: Partly

2. Has the statistical analysis been performed appropriately and rigorously?

Reviewer #1: No

Reviewer #2: Yes

Reviewer #3: N/A

3. Have the authors made all data underlying the findings in their manuscript fully available (please refer to the Data Availability Statement at the start of the manuscript PDF file)?

Reviewer #1: Yes

Reviewer #2: Yes

Reviewer #3: Yes

4. Is the manuscript presented in an intelligible fashion and written in standard English?

Reviewer #1: No

Reviewer #2: Yes

Reviewer #3: No

Reviewer #1: It is an important manuscript describing usability and feasibility of portable CXR instruments. However, there are some comments that require authors' attention.

1. In the background, authors mention, lines 79-80 "..... has adapted the WHO

80 guidelines for use of UPDX for early diagnosis of TB in children 4 years and above and adults..: This needs reference where WHO suggests this as a policy.

2. In the study design section, authors have missed adding the information on when the instrument was received on site and the time that the technicians or others actually had to have real experience before giving the interview. How many days before the interview period (6th April) were the equipment received and were used? This will help in understanding how much experience people had with these portable devices

3. In line 234, it is mentioned the questionnaire were pre-tested. This needs some more explanation on how and who pre-tested it

4. Maybe Table 2 needs a different presentation. It might be a good idea to show it in form of different themes such as questions related to instrument ease, maintenance, image quality etc. Rather than mean, median and SD, it would be good to show frequency in each category for these questions.

5. There was a strong disagreement on the device being portable enough to be carried by one person. This needs some more context in the text with qualitative features.

6. Table 3 does not need all indicators. Would be good to decide which one describes the results best, mean SD or Median IQR or even range.

7. The authors need to mention certain limitations of the manuscript such as: this is only on one device and software combination. We do not know if this can be extrapolated to another software on the same portable device. Additionally, it is important to mention that as the respondents are concerned about the quality of images on this device, there is more evidence needed on the assess the yield and accuracy of these devices with CAD software. Its applicability in the field will be limited if the software cannot yield scores for such low-quality images by the portable CXR devices. These need to be added in the limitations and future considerations.

Reviewer #2: SUMMARY

Using mixed quantitative and qualitative methods, the authors conducted a cross-sectional descriptive study in 2023 to determine the feasibility, acceptability, and ease-of-use of the ultraportable digital X-ray with CAD (UPDX-CAD) in Nigeria. They interviewed 37 HCW and 20 clients (HC attendees). A helpful summary of their findings is made in the Discussion which will doubtless convey to other potential programs in LMICs the benefits and challenges of the UPDX-CAD system. The study is unique, since such a study has not been conducted in other countries.

MAJOR COMMENTS

1. Abstract: It is better to show the key facilitators and barriers in the Abstract rather than dwell on numbers of radiographers, male sex, etc. Suggest combining HCW, clients, KIs.

2. Results: Table 1. Suggest showing education levels by HCW and clients and not combining them since the rest of Table 1 is divided that way.

3. Results: Figure 3 is a very helpful summary. Show all acronyms in the footnote (e.g., TBLS).

4. Discussion “… and also documented the significant potential of the UPDX to improve case detection” seems like an overstatement since the present study did not document any cases. Perhaps the authors intended to state that case detection studies also had HCWs reporting on the ease of use of the UPDX-CAD system.

5. Fig 1. The authors reference a framework for “exploring acceptance and use of UPDX with CAD” but the terms in the figure are not defined. I suggest this figure be changed to a table with definitions shown for each attribute. Otherwise, terms like “efficacy” or “effectiveness” have very different meanings for different types of researchers reading the journal. Initially it was confusing to me that effectiveness or efficacy was studied at all.

6. Discussion. While the authors extensively discuss the viewpoint of HCW, healthcare attendees (or clients) perspectives are not summarized in the Discussion but very briefly mentioned in passing. Please add a summary of the HC attendee element of the framework.

7. Discussion. From my brief research, no other research team has studied facilitators and barriers to UPDX-CAD in Africa. This should be added as a value of this study.

8. Discussion/Conclusion. It would be important for the authors to propose future steps to implement the UPDX-CAD system. Are cost-effectiveness studies needed? Are impact studies needed in Africa such as the one you reference in Peru (https://pubmed.ncbi.nlm.nih.gov/34234000/) or as was done for another new TB diagnostic technology, GeneXpert (https://pubmed.ncbi.nlm.nih.gov/26187490/))?

MINOR COMMENTS

1. Introduction: reference #1 of the authors is not consistent with some of the statistics provided; please recheck. Nigeria was 4.5% of the global TB burden and was 4th highest not 2nd highest in the “global gap”.

2. Methods/Results: correct capitalizations: “likert” (p.15), “microsoft excel” (p.15) “mobisun” (p.22)

3. Methods: “LGBTBLS” (p.15) what does this stand for?

4. Results: since the authors introduce three groups (healthcare workers, key informants, and clients), I suggest that you provide statistics in these three groups.

5. Results: varying verb tenses makes reading paragraph # 1 in the Results ungainly

6. Table 3: first row shows the number of minutes it took to set up UPDX but the statistics use either numbers of minutes or ranges of minutes. It cannot be both. Did the team collect actual numbers of minutes or ask for ranges of minutes?

7. Discussion. Nothing was mentioned about the cost of the system. Although it is somewhat beyond the scope of this study, it would be helpful to know for those of us working in developing country settings with high TB burdens and interested in implementing this system.

Reviewer #3: This is an important study as TB screening using ultra-portable DCXR and CAD is scaled up. I commend the authors for looking at the operational feasibility aspect.

However, the authors need to critically review and re-work all the sections of the submitted paper. Some fundamental aspects that lack clarity include: Which UPDX and CAD software was in use in the 8 states? Was it different across the states or are the respondents all talking about the same model? Is it generalisable to other types?

The introduction can be made more focused and succinct. For example, the authors may consider cutting out the sections line 46-55 as these are general and do not necessarily add to the background of the topic at hand. Also, consider moving a lot of the conceptual framework literature in the methodology and the discussion.

Methodology – Though the study population is clear, there is lack of clarity whether all the 8 provinces were equally represented across the other cadres except radiographers. Data collection process also needs to be refined further for clarity. How many research assistants were there? Line 230, who interviewed the key informants?

Results- Line 274- Clarify what is meant by previous experience with UPDX? Line 292 Table 2 is slightly blurred. Line 300 Table 3- Were the responses triangulated by for example by observation? Or checking registers?

Caution in interpretation for example, Line 324- the authors report 90% agreed the image quality is no different, however from the table only 60% had scores 4/5, 3 respondents were neutral, was that translated to mean agree?

Discussions- The authors can fine tune the discussions further. Also the limitation raised - first to assess feasibility of a new product (new technology) should not be considered as such. Also, the recommendations highlighted new bits of information not captured in the results section.

Overall, the paper has long sentences, some grammatical errors, repetition in some cases and use of abbreviations at first mention and later in the text the full meaning indicated.

**Do you want your identity to be public for this peer review?** For information about this choice, including consent withdrawal, please see our Privacy Policy

Reviewer #1: No

Reviewer #2: **Yes: ** Taraz Samandari

Reviewer #3: No

---

## [Editor Report · Decision Letter 1]

8 Sep 2025

Operational feasibility of the Ultra-Portable digital X-rays with Computer-Aided Detection (CAD) for community active case finding for TB in Nigeria: Health care workers and client’s perspectives

PGPH-D-24-03076R1

Dear Professor EWA,

We are pleased to inform you that your manuscript 'Operational feasibility of the Ultra-Portable digital X-rays with Computer-Aided Detection (CAD) for community active case finding for TB in Nigeria: Health care workers and client’s perspectives' has been provisionally accepted for publication in PLOS Global Public Health.

Best regards,

Sanghyuk S Shin

Academic Editor